# Isolation and Characterization of Yunnan Variants of the Pseudorabies Virus and Their Pathogenicity in Rats

**DOI:** 10.3390/v16020233

**Published:** 2024-02-01

**Authors:** Chunlian Song, Hua Ye, Xue Zhang, Yalun Zhang, Yonghui Li, Jun Yao, Lin Gao, Shanqiang Wang, Yougeng Yu, Xianghua Shu

**Affiliations:** 1College of Animal Medicine, Yunnan Agricultural University, Kunming 650201, China; 2011009@ynau.edu.cn (C.S.); yehua0615abc@163.com (H.Y.); xuezhangx@outlook.com (X.Z.); zylynnydx@163.com (Y.Z.); liyonghui2806@163.com (Y.L.); 2Yunnan Tropical and Subtropical Animal Virus Diseases Laboratory, Yunnan Animal Science & Veterinary Institute, Kunming 650224, China; yaojun_joshua@hotmail.com (J.Y.); 13987123280@163.com (L.G.); 3Weixin County Animal Husbandry Technology Extension Station, Zhaotong 657000, China; wxxxmzh@163.com; 4Animal Disease Prevention and Control Center of Weixin County, Zhaotong 657000, China; yyg018@163.com

**Keywords:** characterization, isolation, mutant strain, rat pathogenicity, swine pseudorabies, Yunnan

## Abstract

Porcine pseudorabies has long existed in China and is a serious threat to the Chinese farming industry. To understand the prevalence and genetic variation of the porcine pseudorabies virus (PRV) and its pathogenicity in Yunnan Province, China, we collected 560 serum samples across seven Yunnan Province regions from 2020 to 2021 and detected anti-gE antibodies in these samples. Sixty-one clinical tissue samples were also collected from pigs with suspected PRV that were vaccinated with Bartha-K61. PRV-gE antibodies were found in 29.6% (166/560) of the serum samples. The PRV positivity rate in clinical tissue samples was 13.1% (8/61). Two isolates, PRV-KM and PRV-QJ, were obtained. The identity of the gB, gD, and gE genes between these isolates and the Chinese mutants exceeded 99.5%. These isolates and the classical Fa strain were used to infect 4-week-old rats intranasally to assess their pathogenicity. All infected rats showed the typical clinical and pathological features of PRV two days post-infection. The viral loads in the organs differed significantly among the infected groups. Viruses were detected in the saliva and feces at 12 h. Significant dynamic changes in total white blood cell counts (WBC), lymphocyte counts (Lym), and neutrophil counts (Gran) occurred in the blood of the infected groups at 24 and 48 h. These results show that mutant PRV strains are prevalent in Bartha-K61-vaccinated pigs in Yunnan Province, China. Moreover, rats shed PRV in their saliva and feces during early infection, indicating the need for rodent control in combatting PRV infections in Yunnan Province, China.

## 1. Introduction

Porcine pseudorabies (PR) is caused by the pseudorabies virus (PRV). PRV can infect a wide range of animals, but pigs are its natural hosts. It has been shown that PRV can still be detected one week after a pig is infected and that, during this process, PRV-infected pigs have the ability to excrete the virus outward, thereby infecting susceptible pigs [1]. This makes it difficult to completely eradicate PRV, seriously threatening the development of the farming industry in many countries, including China. The disease was first reported in cattle in 1813 [2] and became endemic in European swine farms in the 1940s. A highly virulent PRV strain was reported in parts of the United States in the 1960s [3,4]. Highly virulent PRV strains were also reported in China during the same period, which caused significant economic losses to the pig industry [5]. The Bartha-K61 vaccine strain lacks gE, the major virulence factor of PRV, and the detection of gE antibodies in the sera of pigs vaccinated with the Bartha-K61 vaccine strain using the ELISA method is a way of recognizing infection with PRV strains. With the widespread use of the Bartha-K61 PRV vaccine, PR has been largely controlled worldwide.

However, since 2011, PR outbreaks have occurred in pigs immunized with the Bartha-K61 vaccine in several provinces in China. Moreover, the rate of gE antibody positivity in pigs has increased significantly, suggesting that this outbreak is caused by a mutant PRV strain [6,7]. In recent years, PR outbreaks have also occurred in some areas of Yunnan Province in China. Yao et al. [8] monitored PRV in Yunnan Province, China, in recent years, and their results showed that the average PRV-gE positivity rate in pigs was 31.37%, indicating that PRV is still widely prevalent in the region.

PRV is a double-stranded DNA virus with an envelope. This virus belongs to the Orthopoxviridae family and is further categorized in the subfamily of alphaherpesviruses. Also, PRV is the cause of Aujeszky’s disease (AD) [5]. Herpesvirus glycoproteins are present in most membranes of infected cells as well as in the viral particle envelope. These membrane proteins play an important role in viral entry and exit, as well as cell-to-cell transmission. In addition, they have the ability to modulate the immune response and promote syncytium formation. Among the 11 glycoproteins encoded by the PRV, the viral envelope proteins gB and gD play key roles in viral entry, virulence, and immune induction. In addition, gE is an important PRV virulence gene closely associated with viral invasion and spread in the nervous system [1,9]. Mutant PRV strains show the same pathogenic characteristics as their wild-type counterparts, but the sequences of some of their genes (gB, gE, and gD) differ significantly from those of conventional PRV strains; the mutated regions in these isolates can affect PRV gene expression and pathogenicity [10].

Additionally, epidemic PRV strains (mutant strains; represented by HeN1 and HLJ8, among others) show enhanced pathogenicity in mice and pigs. The LD_50_ of these mutant PRV strains is much higher than that of classical strains, such as Fa and Ea. Yang et al. [11] tested the pathogenicity of a PRV variant isolated from Henan Province, China, by comparing it to the classical strain Fa. They found that the Henan PRV variant caused extensive tissue damage and 100% mortality in piglets, while the classical strain Fa only caused weak respiratory symptoms. However, few reports exist on the pathogenicity of PRV mutant strains isolated from pigs in Yunnan Province, China. Hence, the present study investigated the prevalence and isolation of PRV from 560 serum samples and 61 clinical tissue samples collected from selected areas of Yunnan Province in 2020–2021. Meanwhile, the pathogenicity of two PRV mutant strains and the classical virulent strain Fa was studied in rats. The results of this study will provide a scientific basis for the molecular epidemiologic study of PRV and PR prevention and control in Yunnan Province, China.

## 2. Materials and Methods

### 2.1. Sample Sources

In total, 560 blood and 61 tissue (brain, lung, and intestinal) samples were collected from piglets, nursery pigs, and fattening pigs immunized with the Bartha-K61 vaccine from pig farms in Zhaotong, Baoshan, Dehong, Kunming, Qujing, Pu’er, and Chuxiong in Yunnan Province, China, from 2020 to 2021.

### 2.2. Enzyme-Linked Immunosorbent Assay

All serum samples were tested for gE antibody levels using a commercial enzyme-linked immunosorbent assay kit (Wuhan Keqian Biological Co., Ltd., Wuhan, China). To define positivity, *S* was the optical density value of the sample at 630 nm (OD 630 nm), *N* was the average OD 630 nm value of the negative control, and the *S/N* ratio value was calculated. If the sample *S/N* ratio was <0.35, it was considered positive; if the sample *S/N* ratio was >0.4, it was considered negative.

### 2.3. Detection of PRV

Tissue samples were ground and centrifuged for 10 min at 12,000 rpm using a refrigerated high-speed centrifuge (Thermo Fisher, 41,736,095, Waltham, MA, USA) at 4 °C, and the supernatant was collected for DNA extraction (TaKaRa, 9766, Dalian, China). PRV gE primers were designed according to the PRV gene sequence published on GenBank (Accession Number: OR271601.1), and the target gene amplified fragment was 810 bp (Table 1). The upstream primer (F) was derived from sites 124,044–124,063 of the PRV nucleotide sequence, and the downstream primer (R) was derived from sites 124,834–124,853. The extracted DNA was amplified by PCR using gE-specific primers to detect PRV. The PCR reaction conditions were as follows: 95 °C for 5 min and then 35 cycles of 95 °C for 40 s, 58 °C for 40 s, and 72 °C for 60 s, followed by 72 °C for 5 min. After 1% agarose gel electrophoresis verification, the PCR amplicons were cloned into the pMD18-T vector, and the recombinant plasmid was sequenced by Shanghai Sangong Biotechnology Co., Ltd. (Shanghai, China). The sequencing results were confirmed via sequence comparison against PRV sequences in the National Center for Biotechnology Information (NCBI) database.

### 2.4. Virus Isolation

PRV-positive samples were selected for viral isolation. Specifically, the supernatant of the positive homogenate was filtered using a 0.22 µm filter (Biosharp, Beijing, China) and inoculated into a monolayer of BHK-21 cells. The cells were then incubated at 37 °C under 5% CO_2_ to allow adsorption for 1–2 h. The inoculum was discarded, and a maintenance solution (DMEM with 1% fetal bovine serum (Gibco, Waltham, MA, USA) and 1% penicillin–streptomycin antibiotics (Gibco, USA)) was added. Then, the cells were incubated for 3–4 days, and cellular lesions were observed. When 85% cytopathogenic effect (CPE) was observed, the supernatant was collected and used for PCR. The viral isolates were used to calculate the TCID_50_ using the Reed–Muench method [12].

### 2.5. Analysis of Genetic Variations

The primers used to amplify the gB, gD, and gE gene fragments were designed according to the methodology for primer design in Section 2.3, and detailed primer information is provided in Table 1. The whole genome of the extracted PRV isolate was used as a template and was PCR-amplified using the reaction system described in Section 2.3. The PCR products were cloned into the pMD18-T vector, and the constructs were sequenced by Shanghai Sangong Biotechnology Co. The sequencing results were compared with the gene sequences of 15 domestic and foreign PRV strains in GenBank in an evolutionary tree analysis using MEGA 6.0 [13], which involved 1000 bootstrap repetitions (Table 2).

### 2.6. Pathogenicity Test in Rats

Twenty-four specific-pathogen-free male rats (purchased from Kunming Medical University, Yunnan, China) were randomly divided into the uninfected control (UC), PRV-Fa (classical PRV strain; GenBank Accession Number: KM189913.1), PRV-KM, and PRV-QJ groups (PRV-KM and PRV-QJ are the PRV strains isolated in this study). The Fa strain was provided by Yao Jun from the Key Laboratory of Tropical Subtropical Animal Viral Diseases in Yunnan Province. The PRV-positive rat groups were infected with a 100 TCID50 dose of the relevant viral stock solution via nasal drip, while the uninfected control (UC) group was infected with 50 μL of phosphate-buffered saline via nasal drip. All the animal trials in this study were approved by the Animal Ethics Committee of the Yunnan Agricultural University, Kunming, China.

### 2.7. Measurement of Blood Physiological Indices

Four groups of rats were anesthetized at 0, 24, and 48 h after infection, and blood was collected from their orbits using sterile capillary glass tubes; the blood was collected in tubes containing anticoagulant and was analyzed using a fully automated hematology analyzer (COULTER-JT, Beckman Coulter, Brea, CA, USA) in order to determine three indices of blood physiology: white blood cell count (WBC), lymphocyte count (Lym), and neutrophil granulocyte count.

### 2.8. Organ Index

Briefly, 72 h after PRV infection, rats were anesthetized and killed by decapitation, and the brain, heart, liver, spleen, lungs, kidneys, and testes were collected and weighed. The organ index was calculated based on the ratio of organ weight to body weight in rats [14]. All the collected organs were also frozen in liquid nitrogen for viral load calculation and histological analyses.

### 2.9. Viral Load Calculation

Saliva and feces were collected at 12, 24, 36, 48, and 60 h after the rats were infected and were stored at −80 °C until further analysis. Copies of the viral genome were detected in saliva; feces; and seven organs, including the brain, heart, and liver of three groups of infected rats, using a SYBR Green-based real-time quantitative PCR (FQ-PCR) method (Bio-Rad, Hercules, CA, USA) [15]. The total PRV DNA was extracted using a DNA extraction kit (TaKaRa, 9766). The primers for the amplification of the PRV gE gene were designed according to the method of primer design in Section 2.3, and the target gene was amplified at 139 bp (Table 1). The FQ-PCR reaction conditions were as follows: 95 °C for 5 min and then 40 cycles of 95 °C for 30 s, 59.2 °C for 30 s, and 72 °C for30 s, followed by 65 °C for 5 s and 96 °C 5 s. The gene copy number in the reaction is derived by determining the threshold cycle (CQ) value, which corresponds to the CQ value of a serial 10-fold dilution of a standard plasmid. FQ-PCR standard curve: virus copies = −3.679 log Cq + 55.72 (R2 = 0.999).

### 2.10. Histological Analysis

The brain, heart, liver, and other organs of the virus-positive groups were collected and fixed in 10% formaldehyde for pathological tissue sectioning. Formaldehyde-fixed and trimmed tissues were dehydrated, made transparent, dipped in paraffin, embedded and sectioned, baked at 60 °C for 2 h, stained with hematoxylin and eosin (HE), and sealed with neutral gum. Finally, the slides were observed under a light microscope (Chongqing Photoelectric Instrument, Chongqing, China, XDS-1B).

### 2.11. Data Processing and Analysis

Data processing was performed using GraphPad Prism 8 software (Version 10.1., GraphPad Inc., La Jolla, CA, USA). Statistical analyses were performed using one-way ANOVA for inter-group comparisons, with *p* < 0.05 indicating a significant difference, *p* < 0.01 indicating a highly significant difference, and *p* > 0.05 indicating a non-significant difference.

## 3. Results

### 3.1. Epidemiological Analysis of Porcine PRV-gE Serum in Yunnan Province in 2020–2021

Of the 560 porcine serum samples collected, 166 were seropositive for PRV-gE-specific antibodies, with an overall positivity rate of 29.6%. The overall average seropositivity rate for PRV-gE was significantly lower in piglets (17.7%, 20/111) than in nursery pigs (27.6%,61/220) and fattening pigs (36.9%, 85/229) (Figure 1A).

### 3.2. Virus Isolation and PCR Identification

Of the 61 clinical tissue samples collected, 8 were PRV-positive, with a PRV-positive rate of 13.1%. After the positive clinical samples were inoculated into BHK-21 cells, two PRV strains were obtained after isolation and purification, labeled PRV-KM and PRV-QJ. After inoculating the two PRV strains into BHK-21 cells for 3 d, obvious CPE lesions appeared, with rounded cells and multiple cells clustered together; in contrast, normal control cells grew in a uniform monolayer (Figure 1B). Th extracted genomic DNA from each isolate was then used as a template for PCR amplification. The size of the PCR-amplified fragment of the PRV gE gene was approximately 810 bp (Figure 1C).

The isolated viruses were also inoculated into BHK-21 cells and cultured in 96-well plates for 72 h. The cytopathic effect of the PRV strains was determined according to the number of wells with cytopathic lesions. The TCID_50_ values of the PRV-KM and PRV-QJ strains were 10^6^.^75^/mL and 10^6^.^25^/mL, respectively, while that of the classical Fa strain was 10^7^.^13^/mL.

### 3.3. Phylogenetic Analysis

The gB, gD, and gE gene sequences of the PRV isolates were subjected to phylogenetic tree analysis along with the sequences of 15 other representative strains published in GenBank. The results showed that PRV-KM and PRV-QJ, which were closely related to the new epidemic (mutant) strains that became prevalent in China after 2012, were located in the same branch (Figure 1D–F). They were also similar to Asian strains (e.g., the Ea strain) but more distant from the European and American strains (e.g., the NIA3 strain), which were prevalent in China before 2011.

### 3.4. Nucleotide Sequence Identity Analysis of the PRV gB, gD, and gE Genes

The nucleotide sequence identity of the gB, gD, and gE genes of the two PRV isolates in comparison with each other was 99.7%, 99.7%, and 99.6%. In addition, the nucleotide sequence identity of the gB, gD, and gE genes of the two PRV isolates compared with the Chinese PRV variant was 99.6–99.9%, 99.5–100.0%, and 99.5–99.7%. The nucleotide sequence identity of the gB, gD, and gE genes was 99.1–99.4%, 98.7–99.1%, and 99.1–99.2 for the two PRV isolates compared with the Chinese classical PRV strain (Table 3).

### 3.5. Pathogenicity of the PRV-KM and PRV-QJ Isolates and the Classical Fa Strain in Rats

To analyze the pathogenicity of the PRV variants isolated in this study, we infected rats with the two isolated strains and the classical PRV strain Fa. After the infection, the rats showed typical PR symptoms, such as salivation and itching. The organ indices (Figure 2A) of rats in the PRV-positive groups were higher than those in the uninfected control group, and the organs of rats in the positive group were enlarged to different degrees. The postmortem dissection and HE staining of organ sections showed that the PRV-KM, PRV-QJ, and Fa strains caused brain microglial hyperplasia, myocardial swelling and interstitial hemorrhage, hepatocellular necrosis, splenic lymphocyte infiltration, lung congestion, hemorrhage in the glomeruli and glomerular tissue spaces, and testicular interstitial edema and congestion (Figure 2B).

In addition, Figure 2C shows that the viral loads in various organs differed among the mice infected with the PRV-KM and PRV-QJ isolates and the Fa strain, with the highest viral loads observed in the lungs and testes in the Fa group; in the spleen and kidneys in the KM group; and in the brain, heart, and liver in the QJ group. Significant differences in organ viral loads were observed among the groups, except in the spleen and testes. Among the seven organs in rats, the lungs had the highest total viral load, and the testes had the lowest.

According to the results of blood physiological measurements (Figure 2D–F), the total WBC, Lym, and Gran indices of the rats in the three PRV-positive groups showed significant dynamic changes at 24 and 48 h. From 0 to 48 h, WBC and Gran showed an upward trend, while Lym showed a downward trend. However, no significant differences in these indices were observed over time in the uninfected control group.

### 3.6. Dynamic Determination of Detoxification Patterns

To understand the viral shedding dynamics of rats infected with a mutant strain of PRV, oropharyngeal and rectal swabs of rats in the KM and QJ groups were collected at 12 h intervals during the 12–60 h period after infection for PRV viral load determination. Figure 3 shows that the viral loads in saliva and feces during the 12–60 h period have a fluctuating trend, with the salivary viral load in the KM group increasing and then decreasing, and the fecal viral load decreasing and then increasing, while the opposite was true for the QJ group, with the salivary viral load decreasing and then increasing, and the fecal viral load increasing and then decreasing. However, it is undeniable that PRVs were detected in both the saliva and feces of KM and QJ groups at 12 h.

## 4. Discussion

PRV is a widespread, highly pathogenic virus that infects various domestic and wild animals under natural conditions. The virus has mutated considerably in recent years, and the pathogenicity of PRV mutant strains has increased significantly; hence, traditional vaccines can no longer provide complete protection [11]. Currently, the prevention and control of PRV infections are mainly performed using gene deletion vaccines. Developed countries, such as the United States, utilize the gE gene deletion vaccine and accompanying diagnostic methods to control and eradicate PRV infections. However, PRV infections remain prevalent in swine herds with sporadic epidemics in many countries, including China [16,17]. In this study, the average positivity rate of PRV-gE was 29.6%. These results indicate that the PRV epidemic in Yunnan is serious and that the continuous monitoring of PRV infections and understanding the characteristics of the prevalent strains in Yunnan are important for preventing and controlling PR.

Sixty-one cases of suspected PRV infection were observed in this study, of which eight were confirmed to be PRV-positive via PCR, reflecting a positivity rate of 13.1%. This was similar to the national overall nucleic acid-positivity rate for porcine PRV from 2011 to 2021, which is 11.5% [18]. The results of the phylogenetic analysis of the gB, gD, and gE genes of the two PRV strains obtained in this study revealed that these strains belonged to the same branch as the domestic epidemic mutants. This was consistent with the results of the gE and gC gene sequence analysis of five PRV epidemic strains isolated by Gu et al. from four provinces in southeastern China [19]. These results indicate that the two PRV strains isolated in this experiment were mutant strains.

Under the pressure of immune selection, prevalent PRV strains evolve through various mechanisms, such as mutation and deletion; hence, new strains with genetic variations appear. The antigenicity and pathogenicity of these strains may also vary. To further determine the pathogenicity of the PRV isolates, we inoculated 4-week-old rats with the mutant PRV strains KM and QJ and the classical strain Fa. All three groups of infected rats developed typical PR symptoms, such as itching, while histopathological analyses revealed varying degrees of pathological damage to several organs, including the brain, liver, and kidneys. These results are similar to those of studies by Zhou et al. in mice [20]. A previous study [14] showed that PRV infection in mice affects their organs. The organ index and viral load analyses in this study showed that the organs of rats in the infected group were enlarged to different degrees and that the viral loads of the various tissues and organs of rats in the KM, QJ, and Fa groups differed. The highest viral loads were seen in the lungs and testes in the Fa group; the spleen and kidneys in the KM group; and the brain, heart, and liver in the QJ group. Among all the investigated organs, the brain, heart, and liver were found to have the highest viral loads. In the QJ group, the highest viral loads were found in the lungs, brain, heart, and liver. Among the seven organs, the total viral loads of the lungs and brain were higher than those of the other organs, suggesting that different PRV strains might have a preference for different organs and that the lungs and brain may be their main target organs. This result was not consistent with the enhanced pathogenicity observed in common mutant strains. Whether this is related to base substitutions, additions, or deletions in the gB, gD, and gE genes of PRV-KM and PRV-QJ requires further investigation.

Pigs are the only natural hosts of PRV; hence, pigs would be the obvious choice as test animals for evaluating the pathogenicity of PRV isolates. However, due to the limited experimental conditions at the time, we chose to use rats as test animals. Thus, the results of the pathogenicity tests might differ from those that could have been obtained in pigs, which is a major limitation of this study. We plan to use pigs as the test animals for future experiments.

After an animal is infected with a virus, monitoring the viral load in the saliva and feces and the period of detoxification may help understand how the virus spreads and can facilitate the prevention and control of related diseases. In this study, we monitored the viral load in the saliva and feces of rats from 12 to 60 h after infection. The virus was detected in the saliva and feces of rats 12 h after infection, indicating that rats can excrete the virus into the environment within a very short period of time post-infection, leading to the rapid spread of the disease. This might explain why rats can transmit porcine PR.

## 5. Conclusions

In summary, the average serum PRV-gE positivity rate was 29.6% in the samples collected from some areas of Yunnan Province in 2020–2021. The positivity rate for PRV was 13.1%, from which two PRV strains were isolated, named PRV-KM and PRV-QJ, which were identified as mutant strains. Compared to the classical Fa strain, the pathogenicity of these mutant strains to different organs varied. Rats also excreted the virus within a short period post-infection, suggesting that rodent control is crucial for preventing and controlling PR on farms. Collectively, the results of this study provide a theoretical basis for the prevention and control of porcine PR in Yunnan Province, which should be investigated further.

## Figures and Tables

**Figure 1 viruses-16-00233-f001:**
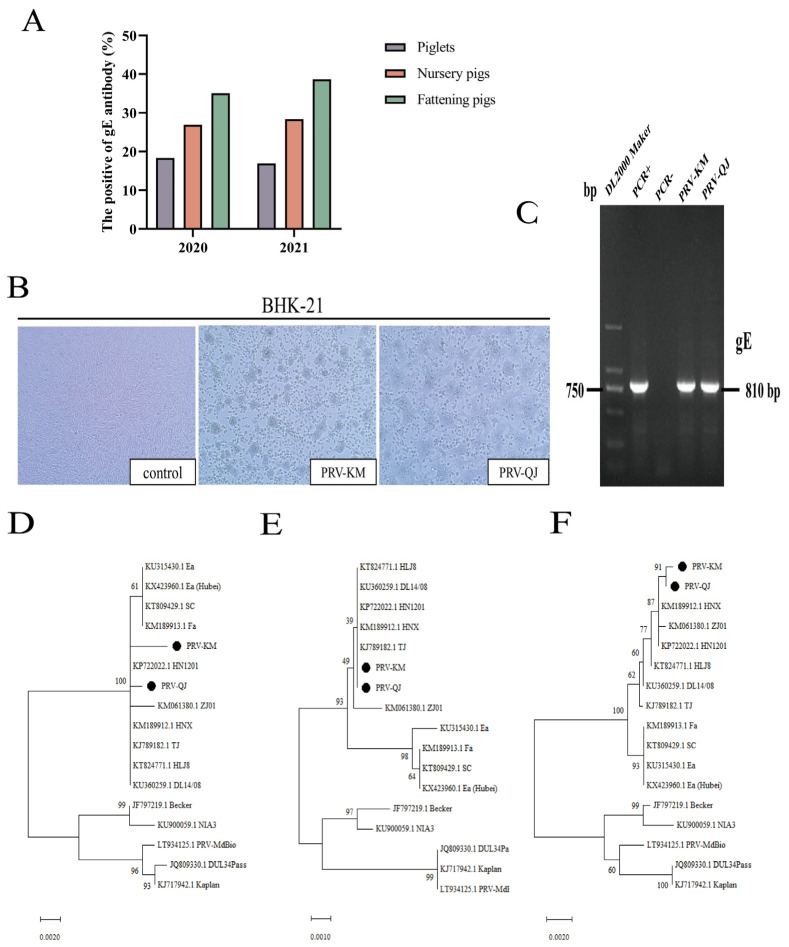
Pseudorabies (PR) epidemiological survey and pseudorabies virus (PRV) isolation: (**A**) serum gE-antibody-positivity rate of different pig herds in seven regions of Yunnan Province during 2020–2021.; (**B**) cytopathogenic effect (CPE) of PRV-KM and PRV-QJ on BHK-21 cells; (**C**) amplification of PRV gE fragments (810 bp) from PRV-KM- and PRV-QJ-inoculated BHK-21 cells. PCR amplification was performed with PCR+ as the positive control and PCR- as the negative control; (**D**) nucleotide phylogenetic tree of the gB gene in PRV isolates; (**E**) nucleotide phylogenetic tree of the gD gene in PRV isolates; (**F**) nucleotide phylogenetic tree of the gE gene in PRV isolates. The black circles indicate the PRV-KM and PRV-QJ strains obtained from this study.

**Figure 2 viruses-16-00233-f002:**
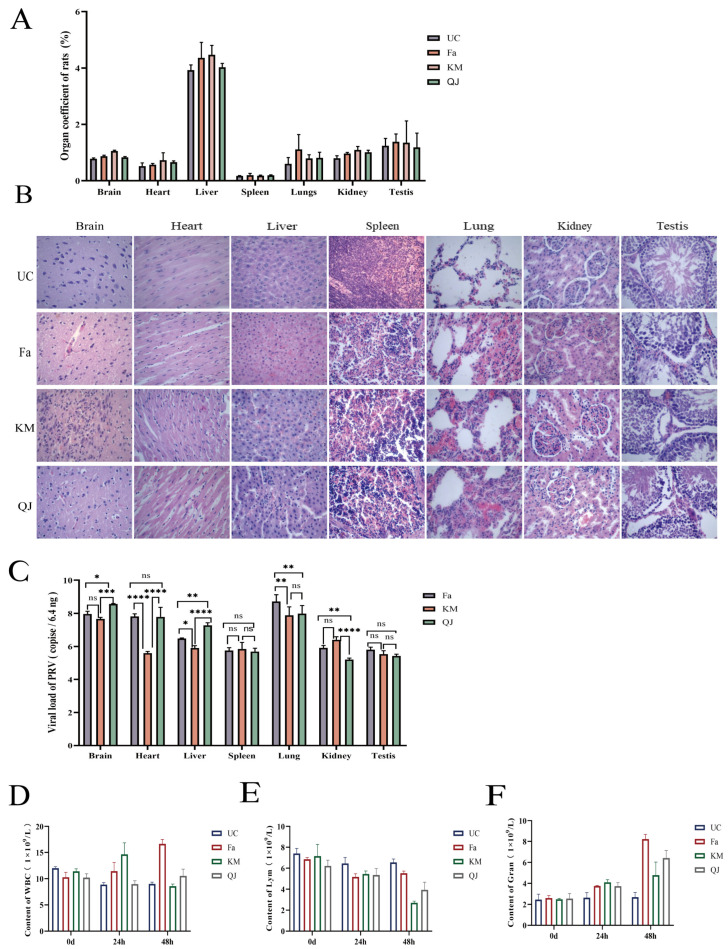
Results of the pathogenicity tests of the pseudorabies virus (PRV) strains Fa, KM, and QJ in rats: (**A**) plot of the results of the uninfected control group (UC) compared with the Fa-, KM-, and QJ-positive organ indices (*n* = 3 in each group); (**B**) pathological changes in mice that died after experimental infection with the Fa, KM, and QJ strains (hematoxylin and eosin staining, 400×); (**C**) plots of the viral load in the brain, heart, liver, spleen, lungs, kidneys, and testes of the Fa, KM, and QJ groups (*n* = 3 in each group). ns means no significant difference, * *p* < 0.05, ** *p* < 0.01, *** *p* < 0.001, and **** *p* < 0.0001; (**D**–**F**) WBC, Lym, and Gran counts in the UC, Fa, KM, and QJ groups at 0, 24, and 48 h post-infection (*n* = 5 in each group).

**Figure 3 viruses-16-00233-f003:**
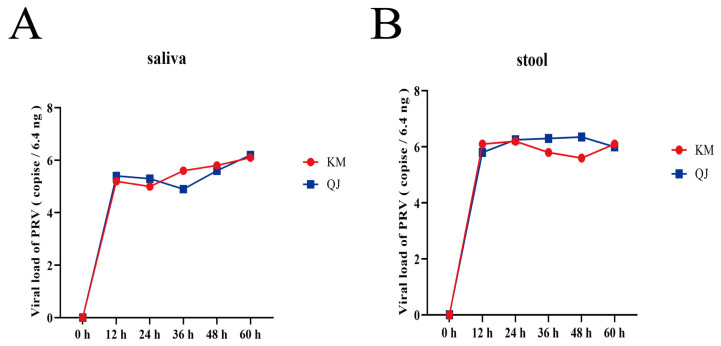
Salivary and fecal viral load results of the PRV-KM and PRV-QJ groups: (**A**) the 0–60-h salivary viral load values of the PRV-KM and PRV-QJ groups (*n* = 3 in each group); (**B**) the 0–60-h fecal viral load values of the PRV-KM and PRV-QJ groups (*n* = 3 in each group).

**Table 1 viruses-16-00233-t001:** Primers specific for PRV-gB, gD, and gE genes.

Virus	Primer Sequence (5′–3′)	Product Length (bp)	Annealing Temperature (°C)
PRV PRV	gE F-CCCAACGACACACGGGGCCTCTA gE R-GCACAGCACGCAGAGCCAGA gB 1F-GCAGTCTTCAGGTCCGTCTTCC	810 893	58.5 58.6
	gB 1R-GGTGTAGGTGTCGTTGGTGGTG		
	gB 2F-CGGCAAGTGCGTCTCCAA	2012	58.5
	gB 2R-GCCCGCTGTTCTTCTTGC		
PRV	gD 1F-CGCGTACCCGTACACCGAGTC	1329	60
	gD 1R-CAGAAACAGCAGCGTCCCGTC		
	gD 2F-GCATCGTCATCATGTGCCTCCAT	867	60.5
	gD 2R-TTGTACGTGCGGTGCTGGTCC		
PRV	gE 1F-GCCACCATCGCAGAGGAACA	1198	58.9
	gE 1R-GGTCATCACGAGCACGTACAGC		
	gE 2F-CCGTCACCGAGGTCCCGAGT	1583	60.5
PRV	gE 2R-CCCATTCGTCACTTCCGGTTTCT gE F-CTACAGCGAGAGAGCGACAACGA gE R-CGACAGCGAGCAGATGACCA GCACAG	139	60

**Table 2 viruses-16-00233-t002:** Sequence information of PRV reference strains.

Number	Strain	GenBank Accession Number	Time	Area
1	TJ	KJ789182.1	2012	China
2	ZJ01	KM061380.1	2012	China
3	HNX	KM189912.1	2012	China
4	HN1201	KP722022.1	2012	China
5	HLJ8	KT824771.1	2013	China
6	DL14/08	KU360259.1	2014	China
7	SC	KT809429.1	1986	China
8	Ea	KU315430.1	1990	China
9	Ea (Hubei)	KX423960.1	1993	China
10	Fa	KM189913.1	2012	China
11	Becker	JF797219.1	2011	USA
12	DUL34Pass	JQ809330.1	2012	Germany
13	Kaplan	KJ717942.1	2011	Hungary
14	NIA3	KU900059.1	2016	UK
15	PRV-MdBio	LT934125.1	2017	Hungary

**Table 3 viruses-16-00233-t003:** Nucleotide sequence identity analysis of the gB, gD, and gE sequences.

	gB	gD	gE
Comparison of two PRV strains obtained in this study	99.7%	99.7%	99.6%
Compared with PRV variants	99.6–99.9%	99.5–100%	99.5–99.7%
Compared with classical PRV strains	99.1–99.4%	98.7–99.1%	99.1–99.2%
Compared with European and American PRV strains	98.3–99.1%	98.2–98.9%	97.4–97.6%

## Data Availability

All data presented in the present study can be found in online repositories.

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
