# Peer review of "Isolation and Characterization of Yunnan Variants of the Pseudorabies Virus and Their Pathogenicity in Rats"

_viruses, 2024, doi:10.3390/v16020233_

Round 1

Reviewer 1 Report

Comments and Suggestions for Authors

Apologies to the authors for submitting a review of a different paper.  This is an interesting paper on PRV that examines the disease in both pigs and rats. Most suggested changes are to add additional details to the paper  

Line 142.  Is there a referenced protocol for this technique?  Why were only male rats used?

Section 2.7.  How was the blood collected?

Section 2.8.  Were rats sedated before decapitation?  What was the end point for rats eg at what point in clinical infection were they euthanized or did they all die on their own? 

Section 2.10   why were only the organs from virus positive groups collected? Weren’t negative controls needed?

Section 3.2   Which 8 tissues were positive and from which group?  This is mentioned in section 3.5 but perhaps it could be briefly mentioned here too.

Line 235. At what day did they die?  In section 2.8 it says they were decapitated. Please clarify

Line 243 “Viral loads ….. in various tissues”  Could you add how they were detected in this paragraph?  It’s described in section 2.4 but sometimes it’s helpful to give the readers a reminder it cell culture vs PCR.

Figure 2 is a valuable part of this paper. Would suggest adding a negative control micrograph to section B

Discussion-The font looks different from the previous parts of the paper

Line 306, 322.  Are rats a good animal model for PRV infection studies?  Maybe just add a sentence or two.   Are rats a better model than mice?

Conclusion is back to the previous font.

Reviewer 2 Report

Comments and Suggestions for Authors

The manuscript summarize the pathogenicity test of PRV isolates, but some modifications would be significant to improve the m.s.

In introduction the very first description of the disease is missing and also the synonym 'Aujeszky's Disease.

In Introduction: "PRV mutants in Yunnan" Did you mean "mutants circulating/endemic/detected in Yunnan"?

In Chapters 2.3 and 2.5: Are the PCR oligos from the literature (in that case, insert references) or designed by the authors. In the latter case describe how they have been designed: how many sequences were aligned etc. What is the actual positions of the primers?

In 2.4 : How the two strains were selected for detailed investigation?

In 2.6: KAM or KM? Share more information about the source of the strains? How old, how ill animals were sampled, what were the symptoms, morbidity/mortality figures in the herd, etc...

In 2.9: Detail the qPCR data analysis: number of replications, preparation of the standard dilutions, threshold settings, etc.

General remark for all figures: i) enlarge them, ii) consider to fragment Fig panels into more Figures, iii) complete the labels: the figures should be understandable without the text. Provide more information (for example: "nucleotide phylogenetic tree" is not enough

In Fig. 1 F : Black circles are missing

In Fig 3: and chapter 3.6: implement parallel measurements , show the mean and +-SD in the graph. Without replicates and statistical analysis the statement that "the 12-60 h viral load (...) showed an increasing trend" is poorly grounded.

Comments on the Quality of English Language

The manuscript is understandable, a check by a native speaker would improve the quality of the text.

Reviewer 3 Report

Comments and Suggestions for Authors

The authors report an interesting study that describes the seroprevalence of pseudorabies virus (PRV) in pigs from Yunnan Province in China. After estimating the seroprevalence, the authors then successfully isolated strains of PRV from two clinically affected pigs that had been vaccinated. The pathogenicity of these isolates in a rat infection model was then examined.

The isolates are important as they were obtained from clinically affected pigs that had been vaccinated for PRV. In this context, I would recommend the authors briefly describe the vaccine and the basis for the serological diagnostics use in the current study.

Overall the study is interesting and the conclusions drawn are supported by the presented data.

Specific comments and suggestions for the authors to consider.

Line 16 suggest revision “560 serum samples from pigs across seven Yunnan Province regions”

Line-20 I would suggest these prevalence estimates be rounded to one decimal place, here and throughout the manuscript.

Line 21 Suggest replacing “homology” with “identity”

The term “homology” is an evolutionary term used to describe characteristics that share a common ancestor. Consequently, it is a “yes” or “no” question, that cannot be quantified. When comparing sequence data, percentage identity is most appropriate for nucleic acids. While percentage identity or percentage similarity are most appropriate for protein sequences.

Line 35 Are the authors able to provide a more specific time period for the time pigs carry PRV? The term “long time” is very subjective.

Line 36 suggest revision to “susceptible pigs”

Line 43-49 – there seems to be something missing in this background text. Reading between the lines, I am assuming that the gE antibody ELISA is able to distinguish between pigs vaccinate with the Bartha-K61 strain and those that have been infected with a “mutant” PRV strain.

Is this correct?

Or is the vaccine strain gE negative?

Suggest adding some text to describe the basis of this test.

Line 52 suggest deletion of “for short”.

Line 70 suggest revision “strains isolated from pigs in Yunnan”

Line 94 These primer sequences could be added to Table 1.

Line 132 suggest revision – “uninfected control (UC)” or “mock infected control (MIC)”

Line 136 suggest superscripting “6” in the dose of virus used. Assuming the infectious dose was 1000000 TCID50 and not 106 TCID50.

Line 136-138 – Please clarify how these doses were administered. The text first suggests the doses were injected, but then suggests intranasal installation was used. As the route and method of challenge can affect the clinical outcome it is important that the method of challenge is clear. Were the challenge doses delivered intranasally by dropwise installation?

Line 166 There are no results presented for immunohistochemistry. 

Line 158 These primer sequences could also be added to Table 1.

Line 191 suggest revision “Extracted genomic DNA from each isolate”

Line 209 Figure 1C – in the gel image the horizontal black lines used to indicate molecular weights should not obstruct the image. 

Line 217 – see previous comment on the term “homology”.

Line 218-223 – I find this text and Table 3 rather uninformative. For sequence comparisons of this type to be informative there has to be a reference point. As an example, in line 2 of Table 3, the strains PRV-KM and PRV-QJ from this study share 99.7% sequence identity with respect to the gB gene, however it is not clear which strain is used as the reference for this estimation. A more conventional way of depicting this type of data is an identity matrix, where each of the strains are compared on a line by line basis. Adding identity matrices for the three genes of interest for all of the isolates to the manuscript would be difficult to justify. Perhaps they could be provided as supplemental files. While Table 3 could be revised to compare the PRV-KM and PRV-QJ from this study to the Fa strain, which of direct relevance to the challenge component of the study. Or alternatively the relevant sequences from the vaccine strain.

Line 225 I think “similarity” should be replaced by “identity”.

Line 263 – I am not sure what the authors mean by “detoxification”, based on the subsequent text I think this term is referring to viral shedding dynamics.

Line 278 Figure 3 – the graphs show “Relative mRNA expression of PRV”. However this is virus detected in saliva and feces, these samples are most likely to contain mature infectious virus and therefore best detected via genomic DNA. Can the authors please clarify how these samples were tested?

Comments on the Quality of English Language

See comments to authors.

Round 2

Reviewer 1 Report

Comments and Suggestions for Authors

Apologies to the authors for submitting a review of a different paper.  This is an interesting paper on PRV that examines the disease in both pigs and rats. Most suggested changes are to add additional details to the paper  

Line 142.  Is there a referenced protocol for this technique?  Why were only male rats used?

Section 2.7.  How was the blood collected?

Section 2.8.  Were rats sedated before decapitation?  What was the end point for rats eg at what point in clinical infection were they euthanized or did they all die on their own? 

Section 2.10   why were only the organs from virus positive groups collected? Weren’t negative controls needed?

Section 3.2   Which 8 tissues were positive and from which group?  This is mentioned in section 3.5 but perhaps it could be briefly mentioned here too.

Line 235. At what day did they die?  In section 2.8 it says they were decapitated. Please clarify

Line 243 “Viral loads ….. in various tissues”  Could you add how they were detected in this paragraph?  It’s described in section 2.4 but sometimes it’s helpful to give the readers a reminder it cell culture vs PCR.

Figure 2 is a valuable part of this paper. Would suggest adding a negative control micrograph to section B

Discussion-The font looks different from the previous parts of the paper

Line 306, 322.  Are rats a good animal model for PRV infection studies?  Maybe just add a sentence or two.   Are rats a better model than mice?

Conclusion is back to the previous font.

Reviewer 2 Report

Comments and Suggestions for Authors

The Authors improved the quality of the manuscript. 

Reviewer 3 Report

Comments and Suggestions for Authors

The authors have adequately addressed the comments and suggestions I made on the submitted version of their manuscript.

I have no further comments.
